# Assessing the risks of treatment in Parkinson disease psychosis: An in-depth analysis

Katherine Longardner[1], Brenton A. Wright[1], Aljoharah Alakkas[2], Hyeri You[3], Ronghui Xu[4], Lin Liu[4], Fatta B. Nahab[1] *

1 Department of Neurosciences, University of California San Diego, La Jolla, California, United States of America, 2 Department of Neurology, Beth Israel Deaconess Medical Center, Boston, Massachusetts, United States of America, 3 Biostatistics Unit, Altman Clinical and Translational Research Institute, University of California San Diego, La Jolla, California, United States of America, 4 Division of Biostatistics and Bioinformatics, Herbert Wertheim School of Public Health and Human Longevity Science, University of California San Diego, La Jolla, California, United States of America

* fnahab@health.ucsd.edu

**Data Availability Statement:** The minimal underlying data set is available on Dryad (DOI: 10. 6076/D1GK51).

**Funding:** This work was supported by ACADIA pharmaceuticals (San Diego, CA) and the National

## Abstract

### Background

Parkinson disease (PD) psychosis (PDP) is a disabling non-motor symptom. Pharmacologic treatment is limited to pimavanserin, quetiapine, and clozapine, which do not worsen parkinsonism. A Food and Drug Administration black box warning exists for antipsychotics, suggesting increased mortality in elderly patients with dementia. However, the reasons for higher mortality are unknown.

### Aim

Expanding on prior work exploring mortality in treated PDP patients, we conducted a retrospective comparison to understand the links between treatment regimen, clinical characteristics, and negative outcomes.

### Methods

Electronic medical record data extraction included clinically diagnosed PD patients between 4/29/16-4/29/19 and excluded patients with primary psychiatric diagnoses or atypical parkinsonism. Mortality and clinical characteristics during the study period were compared between untreated patients and those receiving pimavanserin, quetiapine, or both agents (combination). Mortality analyses were adjusted for age, sex, levodopa equivalent daily dose (LEDD), and dementia.

### Results

The pimavanserin group (n = 34) had lower mortality than the untreated group (n = 66) (odds ratio = 0.171, 95% confidence interval: 0.025–0.676, *p* = 0.026). The untreated group had similar mortality compared to the quetiapine (n = 147) and combination (n = 68) groups. All treated groups had a higher LEDD compared to the untreated group, but no other

Institutes of Health (NIH) (University of California San Diego Clinical and Translational Science Award grant number UL1TR001442). The content is solely the responsibility of the authors and does not necessarily represent the official views of the NIH. The funding providers had no role in study design, data collection and analysis, decision to publish, or preparation of the manuscript.

**Competing interests:** The authors have declared that no competing interests exist.

differences in demographics, hospitalizations, medical comorbidities, medications, or laboratory values were found between the untreated and treated groups.

## Conclusions

PDP patients receiving pimavanserin had lower mortality than untreated patients. We found no other clear differences in clinical characteristics to explain the mortality risk. Prospective randomized trials are needed to definitively identify the optimal PDP treatment regimen and associated risks.

## Introduction

Psychosis is a common non-motor symptom in Parkinson disease (PD), with overall prevalence ranging from 26–60%, depending on which symptoms are included [1–3]. Manifestations occur on a spectrum, ranging from a false sense of presence and illusions to formed visual hallucinations and delusions, which can occur with or without insight [4]. PD psychosis (PDP) prevalence increases with age, cognitive dysfunction, disease duration, and dopaminergic therapy. Patients with PDP have higher rates of healthcare utilization, institutionalization, and mortality than those without psychosis [5].

Direct PDP treatment is limited to the few antipsychotic medications that have low affinity for dopaminergic D2 receptors to avoid worsening parkinsonian symptoms; these include quetiapine and clozapine, which have traditionally been used to treat PDP. [6]. Quetiapine is commonly prescribed due to its favorable side effect profile and cost, but its efficacy as an antipsychotic in PD has demonstrated inconsistent results [7]. Clozapine is used effectively in PDP patients who do not respond to other antipsychotics, but its use is limited by the risk of agranulocytosis and need for frequent blood monitoring for this rare side effect. The Food and Drug Administration (FDA) approved pimavanserin on April 29, 2016 as the first agent indicated specifically for treatment of PDP. Pimavanserin does not worsen parkinsonian symptoms due to its unique mechanism of action as a pure 5HT2-A receptor inverse agonist that lacks any activity at dopamine receptors. However, the FDA issued a black box warning for all antipsychotic medications (including pimavanserin, despite its unique mechanism of action) in elderly patients with dementia, suggesting that this medication class is associated with increased morbidity and mortality. Antipsychotic exposure has also been associated with increased mortality risk in PD, compared to people with PD who are not exposed to antipsychotics [7, 8].

We previously reported in a retrospective study that among 676 PD patients treated for psychosis, those receiving pimavanserin had lower mortality than those receiving quetiapine or combination therapy with pimavanserin and quetiapine [9]. However, given that our prior work had not included person-level review, the factors contributing to these mortality differences were unknown. Larger observational cohort studies have found varying results regarding pimavanserin's association with mortality in PD. One study demonstrated among people not residing in long-term care facilities, pimavanserin users had decreased mortality compared to users of other antipsychotics [10]. However, another study found among residents of long-term care facilities that pimavanserin users had higher mortality compared to non-users [11]. In the present study, we aimed to better understand the impact and predictors of mortality in PDP by expanding the study period, conducting an in-depth retrospective review to explore various demographic, clinical, and iatrogenic factors, and controlling for potential confounds with the ability to gather information by individual chart review. We also investigated the

differences in these factors between those treated with antipsychotic agents and PDP patients who remained untreated.

## Methods

This research project was approved by the University of California San Diego (UCSD) Institutional Review Board (Project #190625), and it is conformed to the provisions of the Declaration of Helsinki. We extracted identified patient data from the Epic electronic medical record system (Verona, WI). All patients included were: 1) clinically diagnosed with PD (using International Classification of Diseases, Tenth Revision, Clinical Modification (ICD-10-CM) code), and 2) evaluated at any UCSD Health System facility for any cause between April 29, 2016 and April 29, 2019. Psychosis was diagnosed based on ICD-10 code and antipsychotic medication prescription. For patients prescribed antipsychotic medications, individual chart review was performed to ascertain that the medication was prescribed for treatment of psychosis (i.e., rather than for sleep or mood). We excluded patients with primary psychiatric diagnoses (including bipolar disorder, schizophrenia, schizotypal disorder, and depression with psychotic features), since these "may have drug-induced or tardive parkinsonism related to antipsychotic medication use. We excluded patients with atypical parkinsonism (e.g., multiple system atrophy, progressive supranuclear palsy, drug-induced parkinsonism, vascular parkinsonism), since pimavanserin is only FDA-approved for use in people with PDP. Persons with PDP were categorized according to treatment status: none (untreated), pimavanserin, quetiapine, or both pimavanserin and quetiapine (combination). We excluded patients treated with clozapine monotherapy (n = 2) or clozapine combined with other antipsychotics (n = 9) from the analyses due to low sample size. We did not include other atypical antipsychotic medications, e.g., risperidone or olanzapine, in our search query since these and other antipsychotics with D2 dopaminergic blocking mechanism of action are generally avoided in PD given their propensity to exacerbate parkinsonian symptoms.

We included the following variables in our data query: living/deceased status; demographics (age, sex, payor status, race/ethnicity); Montreal Cognitive Assessment (MoCA) total scores (range 0–30, lower is worse) [12]; Movement Disorders Society Unified Parkinson Disease Rating Scale (MDS-UPDRS) Total Part III scores (range 0–132, higher is worse) [13]; electrocardiogram QTc interval; levodopa equivalent daily dose; frequency and duration of hospital admissions in the UCSD Health System during the study period; comorbidities determined by ICD-10 code (parkinsonian non-motor symptoms including dementia, and common general medical conditions, which included cardiovascular disease, chronic kidney disease, diabetes mellitus type 2, hypertension, hyperlipidemia, and hypercoagulability); medication exposure (medication categories included antipsychotics, antiparkinsonian, cognition-enhancing, antihypotensives, antidepressants, antithrombotics, QTc-prolonging, and potassium-depleting); and laboratory values when available to provide objective data (including complete blood count, chemistry panel, liver enzymes, lipid profile, coagulation profile). For medical exposure/condition variables, the variable was defined as any exposure during the study period. If multiple measurements were available for continuous variables, the values were aggregated and averaged for each patient in the between-group comparisons, these include UPDRS Part III scores, MoCA scores, QTc interval, and laboratory values.

Medication exposure was determined according to whether patients were ever prescribed the antipsychotic medication during the study interval. The combination group included patients that had exposure to both pimavanserin and quetiapine during the study period, but these two agents were not necessarily taken at the same time. Patients who were exposed to antipsychotic medications and discontinued them before the study period were included in

the untreated group. Individual chart reviews were performed for all individuals prescribed antipsychotics to confirm that they took the medication(s) during the study period. Those who did not start the medications prescribed were assigned to the untreated group. We also distinguished between persons that received quetiapine chronically in the outpatient setting and those that received quetiapine only during an inpatient setting (i.e., emergency room visit or hospital admission) and excluded the latter group (n = 34). Individuals without psychosis taking quetiapine in the outpatient setting for sleep or mood were also excluded from analyses (n = 8). Quetiapine doses at the first and last visit during the study period were calculated based on chart review.

Since there is some evidence that dopaminergic agents may increase risk of PDP [14, 15], we calculated levodopa equivalent daily doses (LEDD) at the first and last visits during the study period were using the conversion formula described by Tomlinson et al. [16]. Extended release carbidopa/levodopa was converted using the method described by Hauser, in accordance with the product package insert [17]. First and last visit LEDD were only compared among those with more than one visit. If only one visit occurred during the study period, LEDD was listed under the last visit.

## Statistical analyses

We compared demographics, clinical factors, and medication use between the untreated group and each of the three treatment groups (quetiapine, pimavanserin, and combination) using pairwise univariate tests. For continuous variables, a two-sample t-test was used, while for categorical variables the Chi-square test was applied to each of the three pairwise between-group comparisons. When interpreting the significance of a test, multiple comparison adjustment with Bonferroni correction was applied for the three pairwise treatment group comparisons ($p<0.05/3$ was considered significant). Logistic regression was conducted to compare mortality rates in groups with PDP receiving quetiapine, pimavanserin, or combination therapy to the untreated group. Multivariable logistic regression was used to adjust for age, sex, last visit LEDD, and dementia diagnosis.

In the subgroup of patients taking quetiapine monotherapy in the outpatient setting, we investigated whether a quetiapine dose effect was associated with the following five outcomes: mortality, hospital admission frequency, hospitalization duration, presence of orthostatic hypotension (OH), and average QTc interval. Quetiapine doses were analyzed both as 1) continuous variables and 2) binary variables, separating groups into those receiving less than 50mg of quetiapine daily and those receiving 50mg or greater of quetiapine daily. Linear and logistic regression were used for analyzing the five outcomes as appropriate, and multivariable regression analyses were performed to adjust for age and last visit LEDD.

Analyses were performed using R statistical software version 4.0.2 (R Foundation for Statistical Computing, Vienna, Austria) [18].

## Results

Using our inclusion/exclusion criteria, the sample included 2,994 PD patients– 352 (11.8%) with psychosis. Of these 352 PDP patients, 66 (18.8%) were untreated (did not receive antipsychotics), 34 (9.7%) received pimavanserin, 147 (41.8%) received quetiapine in the outpatient setting, and 68 (19.3%) received combination therapy, thus 315 patients were included in the analyses.

### Mortality

Group mortality rates in our cohort were: 24.2% (untreated), 5.9% (pimavanserin), 20.7% (quetiapine), and 17.1% (combination therapy). The likelihood of mortality was lower in

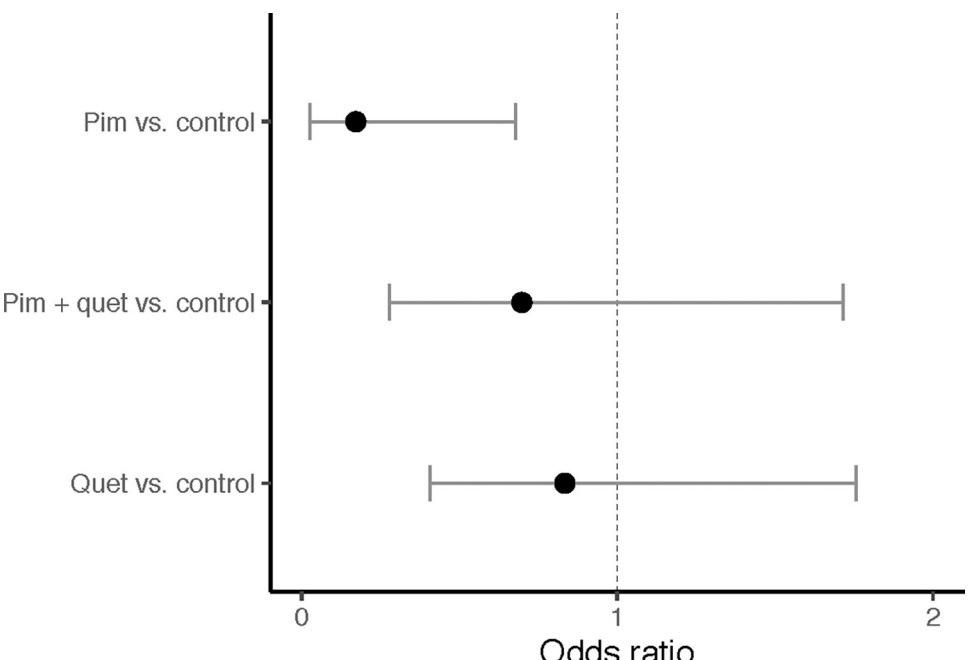

**Fig 1. Mortality odds ratio in untreated and treated groups with Parkinson disease psychosis.** Comparison of mortality odds ratios between Parkinson disease psychosis (PDP) patients not receiving antipsychotics (untreated) and PDP patients treated with either pimavanserin (Pim), quetiapine (Quet), or combination pimavanserin and quetiapine (Pim+Quet), after adjusting for age, sex, last visit levodopa equivalent daily dose, and dementia.

patients receiving pimavanserin compared to untreated patients [odds ratio (OR): 0.195; 95% confidence interval (CI): 0.030, 0.748; $p$ = 0.037], and remained lower after adjusting for age, sex, last visit LEDD, and dementia diagnosis [OR: 0.171; 95% CI: 0.025, 0.676; $p$ = 0.026]. Compared to the untreated group, there were no differences in adjusted mortality for patients receiving quetiapine [OR: 0.833; 95% CI: 0.405, 1.756; $p$ = 0.624] or those on combination therapy [OR: 0.697; 95% CI: 0.277, 1.716; $p$ = 0.433] (Fig 1).

## Demographic data and clinical features

We found no differences in age, sex, race/ethnicity, or payor status between untreated and treated groups. A subset of the patients had data collected for the MDS-UPDRS Part III, MoCA, and QTc interval. Compared to the untreated group's MDS-UPDRS Part III scores (mean 29.0 points, standard deviation (SD) 13.9), motor performance was worse in the quetiapine group (mean 40.6, SD 18.9, $p$ = 0.007) and combination group (mean 41.7, SD 19.1, $p$ = 0.013). Mean MoCA scores were lower in the combination group (16.0 points, SD 7.0) compared to the untreated group (21.2, SD 5.9, $p$ = 0.009). MoCA scores in the combination therapy group trended lower compared to the untreated group, but did not meet statistical significance (mean 17.50, SD 7.20, $p$ = 0.033). There were no differences in mean QTc interval between the treated and untreated groups. Among patients taking dopaminergic medications, the combination therapy group a higher first visit LEDD compared to the untreated group, but this was not statistically significance after multiple comparison correction. Compared with the untreated group, all treated groups had higher LEDD at the last visit. Table 1 shows demographic information and clinical characteristics at baseline (first study visit assessed).

**Table 1. Comparison of demographic and clinical characteristics at baseline between Parkinson disease psychosis (PDP) group not receiving antipsychotics (untreated) and PDP groups treated with pimavanserin, quetiapine, or combination pimavanserin and quetiapine.**

| | Untreated PDP (n = 66) | Pimavanserin (n = 34) | *p*-value | Quetiapine (outpatient) (n = 147) | *p*-value | Combination (n = 68) | *p*-value |
|---|---|---|---|---|---|---|---|
| Age, years mean (SD) | 77.7 (9.3) | 80.2 (6.5) | 0.127 | 76.9 (9.2) | 0.536 | 75.9 (8.8) | 0.238 |
| Sex, female n (%) | 28 (42.4) | 18 (52.9) | 0.431 | 50 (34.0) | 0.306 | 24 (35.3) | 0.503 |
| RACE/ETHNICITY n (%) | | | | | | | |
| White/Caucasian (non-Hispanic) | 55 (83.3) | 25 (73.5) | 0.494 | 117 (79.6) | 0.376 | 56 (82.4) | 0.912 |
| Hispanic/Latino | 6 (9.1) | 6 (15.0) | | 20 (13.6) | | 7 (10.3) | |
| Other | 1 (1.5) | n/a | | n/a | | 2 (2.9) | |
| Unknown | 4 (6.1) | 3 (8.8) | | 10 (6.8) | | 3 (4.4) | |
| PAYOR STATUS n (%) | | | | | | | |
| Government payor | 51 (77.3) | 24 (70.6) | 0.740 | 90 (61.2) | 0.108 | 57 (83.8) | 0.368 |
| Private payor | 13 (19.7) | 9 (26.5) | | 52 (35.4) | | 11 (16.2) | |
| Other | 2 (3.0) | 1 (2.9) | | 5 (3.4) | | n/a | |
| CLINICAL FEATURES mean (SD) | | | | | | | |
| MDS-UPDRS Part III score | n = 25 | n = 14 | 0.125 | n = 38 | **0.007***| n = 22 | **0.013*** |
| | 28.96 (13.92) | 37.64 (17.54) | | 40.63 (18.93) | | 41.73 (19.05) | |
| MoCA score | n = 20 | n = 17 | 0.075 | n = 46 | 0.033 | n = 26 | **0.009*** |
| | 21.20 (5.85) | 17.06 (7.52) | | 17.50 (7.20) | | 16.00 (6.99) | |
| QTc interval | n = 28 | n = 8 | 0.865 | n = 54 | 0.360 | n = 29 | 0.207 |
| | 457.13 (29.93) | 460.23 (47.22) | | 450.49 (32.83) | | 446.01 (35.66) | |
| First visit LEDD | 346.7 (433.2) | 417.8 (350.7) | 0.379 | 348.6 (318.9) | 0.975 | 526.4 (503.3) | **0.028** |
| Last visit LEDD | 407.1 (511.0) | 689.7 (538.7) | **0.014*** | 635.8 (519.9) | **0.003*** | 803.9 (609.1) | **< 0.001*** |

Comparison of age, sex, ethnic/racial group, payor status, motor performance, cognitive assessments, QTc interval, and levodopa daily equivalent dose between Parkinson disease psychosis (PDP) patients not receiving antipsychotics (untreated) and PDP patients treated with either pimavanserin, quetiapine, or combination pimavanserin and quetiapine. *P*-values are unadjusted and shown for each treated group compared to untreated.

* represents *p*-values meeting Bonferroni corrected significance level ($p < 0.016$).

Abbreviations: LEDD: Levodopa Daily Equivalent Dose; MDS-UPDRS: Movement Disorders Society-Unified Parkinson's Disease Rating Scale; MoCA: Montreal Cognitive Assessment; SD: Standard Deviation.

## Hospitalization data

The percentage of individuals hospitalized during the study period was lower in the pimavanserin group compared with the untreated group (20.6% vs. 50%, $p = 0.009$) and was similar between the untreated group and groups receiving quetiapine (39.5%, $p = 0.197$) and combination therapy (39.7%, $p = 0.306$). Hospitalization frequency (number of hospitalizations per individual) was similar between untreated patients and all treated groups. Patients treated with quetiapine trended toward a longer hospital duration compared to untreated patients (mean duration 3.88 days vs. 2.32 days, $p = 0.040$), but after adjusting for multiple comparisons, this difference was not significant. Hospitalization data is shown in Table 2.

## Prevalence of parkinsonian non-motor symptoms

The untreated PDP group had a higher prevalence of mild cognitive impairment (MCI) (39.4% vs. 18.4%, $p = 0.002$), mood disorders (63.6% vs. 40.8%, $p = 0.003$), and urinary symptoms (39.4% vs. 21.8%, $p = 0.012$) than groups who received quetiapine. There was also a trend

**Table 2. Comparison of hospitalization data between Parkinson disease psychosis (PDP) group not receiving antipsychotics (untreated) and PDP groups treated with pimavanserin, quetiapine, or combination pimavanserin and quetiapine.**

| | Untreated PDP (n = 66) | Pimavanserin (n = 34) | p-value | Quetiapine (outpatient) (n = 147) | p-value | Combination (n = 68) | p-value |
|---|---|---|---|---|---|---|---|
| **Patients hospitalized overnight n (%)** | 33 (50.0) | 7 (20.6) | **0.009**[*] | 58 (39.4) | 0.197 | 27 (39.7) | 0.306 |
| **Admissions per patient mean (SD)** | 2.5 (4.3) | 1.6 (1.9) | 0.151 | 1.8 (2.1) | 0.182 | 2.1 (2.2) | 0.522 |
| **Hospitalization duration in days mean (SD)** | 2.3 (2.7) | 2.1 (2.0) | 0.826 | 3.9 (4.5) | 0.040 | 4.1 (6.8) | 0.214 |

Comparison of hospital admission frequency and duration between Parkinson disease psychosis (PDP) not receiving antipsychotic medications (untreated) and PDP patients treated with either pimavanserin, quetiapine, or combination pimavanserin and quetiapine. *P*-values are unadjusted and shown for each treated group compared to untreated. Significance vs. untreated at Bonferroni-corrected $p < 0.016$ is marked with [*].

toward higher prevalence of orthostatic hypotension in the untreated group compared with the quetiapine-treated group (42.4% vs. 25.2%, $p = 0.018$), that did not reach significance after adjusting for multiple comparisons.

## General medical conditions

The untreated group had a higher prevalence of hypercoagulability compared to the quetiapine group (16.7% vs. 4.8%, $p = 0.009$). Compared to the combination group, the untreated group had a higher prevalence of hypertension (62.1% vs. 35.3%, $p = 0.003$) and chronic kidney disease (22.7% vs. 5.9%, $p = 0.011$). No other between-group differences were found.

## Medication exposure

No differences were found in medication exposure by category between treated and untreated groups.

**Quetiapine dose.** While pimavanserin has essentially only one dose (34mg daily), quetiapine is used in a wide range of doses and frequencies. We explored whether quetiapine dose predicted clinical outcomes. Among the 147 individuals that received quetiapine monotherapy for psychosis in the outpatient setting, we evaluated whether quetiapine dose at the first or last visit during the study period was associated with mortality, hospital admission frequency, hospitalization duration, presence of orthostatic hypotension (OH), and QTc interval. Quetiapine doses ranged from 12.5mg to 200mg; 98 patients took less than 50mg daily, and 49 patients took 50mg or greater daily. We found no significant correlations between quetiapine dose and these outcomes in uni- or multivariate analyses. Pimavanserin dose was not compared between groups given its narrow dose window ranging from 10-34mg daily.

## Laboratory values

Serum LDL was lower in the pimavanserin group compared to the untreated group (mean 70.0 mg/dL, SD 6.4, vs. mean 98.0 mg/dL, SD; $p = 0.009$). There were no other significant differences in serum lab values between untreated and treated groups.

## Discussion

Little is known about the mechanisms underlying the safety concerns of antipsychotics broadly, or for quetiapine and pimavanserin in the context of PDP. In this retrospective study, we performed a comprehensive medical record review to replicate our earlier work showing

increased mortality in individuals with PDP receiving quetiapine and no increase in those treated with pimavanserin [9]. In addition, this more in-depth retrospective review explored various demographic, clinical, and iatrogenic factors in those treated with antipsychotic agents and individuals with PDP who remained untreated. Although we expanded the exposure period from two years to three years compared to our previous study, we refined our criteria to exclude patients who had primary psychiatric diagnoses or those who received quetiapine exclusively in the inpatient setting. We did replicate our previous finding that those with PDP who received pimavanserin had lower mortality than untreated individuals, while finding those receiving quetiapine and combination therapy had similar mortality compared to the untreated group after adjusting for age, sex, last visit LEDD, and presence of dementia.

We then explored potential predictors of mortality within this population. Across the various treatment regimens, we found no disparities in demographics, socioeconomic factors, or hospitalizations to explain differences in morbidity or mortality. Groups treated with quetiapine and combination therapy had worse parkinsonian motor symptoms; otherwise, there were no differences in clinical features between groups. We compared PD non-motor symptoms between groups, since mood and sleep disorders, cognitive impairment, and dysautonomia have been associated with decreased survival in PD [19–21]. The untreated group had more frequent urinary symptoms and mild cognitive impairment compared to the those treated with quetiapine but no other non-motor symptom differences were found between groups We also investigated non-PD-related medical comorbidities between groups. The untreated group had an increased prevalence of hypertension and cardiovascular disease (diagnosed by ICD-10 code) compared to the group receiving combination therapy–these comorbidities may also impact life-expectancy. Alternatively, it is possible that individuals taking higher doses of dopaminergic therapy and/or quetiapine may have had lower blood pressure as a side effect of these medications. Finally, we examined exposure to clinically relevant medications that may influence morbidity and mortality. Overall, we observed no differences in medication exposures between the groups. The pimvanserin group had lower serum LDL compared with the untreated group, but no other group differences were found in the subset of individuals with lab data.

In summary, we found no definitive associations in clinical characteristics or comorbidities that accounted for the mortality differences between groups. However, we found several factors that were significant before adjusting for multiple-group comparisons. These included longer hospitalization duration and less prevalent orthostatic hypotension in the quetiapine compared to the untreated group, and lower MoCA scores and higher first visit LEDD in the combination group compared to the untreated group. The exploratory nature of this study limits the conclusions and emphasizes the need for prospective hypothesis-driven studies with larger samples to definitively address whether these findings can be replicated. To date, no study has directly compared pimavanserin to quetiapine for treating PDP. Prospective, longitudinal, randomized controlled clinical trials are needed to definitively identify the optimal treatment regimen and associated risks and guide clinical decision making. Transition to use of one antipsychotic agent over another should be based on a more favorable benefit:risk ratio. Our data suggest that individuals with PDP receiving quetiapine may be incurring additional risks without proven benefits.

Other larger observational studies on pimavanserin's effect on mortality in PDP patients using antipsychotics show mixed results. A three-year retrospective study by Mosholder et al. showed that in Medicare beneficiaries with PD initiating antipsychotic treatment, pimavanserin users (n = 3,227) had lower mortality than atypical antipsychotic users (n = 18,442) during the first 180 days of treatment [10]. However, this association with lower mortality was only found in people residing in the community, not in nursing home residents. Hwang et al.

published a retrospective cohort study of adults 65 years and older residing in Medicare-certified long-term care facilities with PD who were followed for 38 months, showing that pimavanserin users (n = 2,186) had increased mortality compared to non-users (n = 18,212) at three, six, and 12 months after initating pimavanserin [11]. Pimavanserin users also had increased hospitalization rates compared to non-users at one month, but not three months after its initiation. These studies concur that pimavanserin may not offer any mortality benefit in people residing in long-term care facilities. However, these studies did not determine any other clinical differences to account for the mortality differences. Disparate study populations may explain these varied results. Compared with Hwang and colleagues' study population, our cohort, included patients seen in the ambulatory setting. Furthermore, Hwang and colleagues' study population included patients with primary psychiatric diagnoses (i.e., bipolar disorder, schizophrenia) and patients taking other antipsychotic medications. Their population may have included people with drug-induced parkinsonism, and overall likely had worse functional status and more medical comorbidities since their data was gathered from people residing in long-term care facilities.

The strengths of our retrospective electronic medical record-dependent design include the ability for detailed individual chart review to accurately characterize the PDP cohort and accurately confirm the use of antipyschotic medications. The challenge in such study designs is the limited ability to accurately characterize disease duration within the large dataset of patients. Similarly, data regarding initiation and duration of antipsychotic treatment, duration of follow-up, frequency of follow-up, and the provider that diagnosed PD and PDP was not available given the limitations from the medical record system data extraction, since data was not collected in a standardized manner. Given these limitations, we were unable to account for adjustments in antipsychotic medication, changes in antipsychotic agents, or duration of exposure to antipsychotic medications, and thus, could not use survival analysis methodology to analyze mortality since we could not define the exposure start date. For the same reasons, we could not include time-varying covariates. Hospitalization data was limited to those within our academic medical center health system, thus, may underestimate the number of hospital admissions. Similarly, mortality was identified by chart data extraction during the specified time period, and thus, deaths may have been missed in patients with limited follow-up. Despite these limitations, there is no expectation that these variables would differ between the groups. For much of the data presented (e.g., non-motor symptoms, medications, etc.) it is not known whether these first occurred before or after initiating antipsychotic medications due to the methodology of data extraction. It is possible that among the 2994 PD patients in our dataset, the number of untreated PDP patients may have been underrepresented. We acknowledge the small sample size as a limitation to the generalizability of our findings. Additionally, given the restrospective study design, we could not adjust for severity of PDP symptoms or delineate between manifestations of psychosis in our cohort (e.g., hallucinations vs. delusions) or the presence or absence of insight, and we could not determine why one antipsychotic agent was chosen over another. Thus, we acknowledge that between-group differences may exist in psychosis severity or symptoms that could affect whether antipsychotic medication was prescribed, or which agent was prescribed.

Taken together, this expanded retrospective study extends our previous work and again demonstrates that individuals with PDP receiving pimavanserin had lower mortality risk than those who were untreated. While we found no definitive risk factor(s) between these groups to explain the difference in mortality, several notable risk factors should be weighed, mitigated, and monitored by a prescribing practitioner managing patients with PDP, including: parkinsonian non-motor symptoms such as cognitive impairment, cardiovascular comorbidities, and polypharmacy.

## Author Contributions

**Conceptualization:** Katherine Longardner, Fatta B. Nahab.

**Data curation:** Katherine Longardner, Brenton A. Wright, Aljoharah Alakkas, Fatta B. Nahab.

**Formal analysis:** Hyeri You, Ronghui Xu, Lin Liu.

**Investigation:** Katherine Longardner.

**Methodology:** Brenton A. Wright, Hyeri You, Ronghui Xu, Lin Liu, Fatta B. Nahab.

**Supervision:** Ronghui Xu, Lin Liu, Fatta B. Nahab.

**Writing – original draft:** Katherine Longardner, Fatta B. Nahab.

**Writing – review & editing:** Brenton A. Wright, Aljoharah Alakkas, Hyeri You, Ronghui Xu, Lin Liu, Fatta B. Nahab.

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
