## [Decision Letter · Decision Letter 0]

19 Sep 2022

PONE-D-22-22035Assessing the risks of treatment in Parkinson disease psychosis: an in-depth analysisPLOS ONE

Dear Dr. Nahab,

Thank you for submitting your manuscript to PLOS ONE. After careful consideration, we feel that it has merit. Therefore, we invite you to submit a revised version of the manuscript that addresses the points raised during the review process.

We look forward to receiving your revised manuscript.

Kind regards,

Antonina Luca, MD, PhD

Academic Editor

PLOS ONE

Journal Requirements:

"This work was supported by ACADIA pharmaceuticals (San Diego, CA) and the National Institutes of Health (NIH) (University of California San Diego Clinical and Translational Science Award grant number UL1TR001442). The content is solely the responsibility of the authors and does not necessarily represent the official views of the NIH."

"This work was supported in part by ACADIA pharmaceuticals (San Diego, CA) and the National Institutes of Health (NIH) (University of California San Diego Clinical and Translational Science Award grant number UL1TR001442). 

The content is solely the responsibility of the authors and does not necessarily represent the official views of the NIH.

The funding providers had no role in study design, data collection and analysis, decision to publish, or preparation of the manuscript."

Reviewers' comments:

Reviewer's Responses to Questions

**Comments to the Author**

1. Is the manuscript technically sound, and do the data support the conclusions?

Reviewer #1: Yes

Reviewer #2: Yes

2. Has the statistical analysis been performed appropriately and rigorously? 

Reviewer #1: Yes

Reviewer #2: Yes

3. Have the authors made all data underlying the findings in their manuscript fully available?

Reviewer #1: Yes

Reviewer #2: Yes

4. Is the manuscript presented in an intelligible fashion and written in standard English?

Reviewer #1: Yes

Reviewer #2: Yes

5. Review Comments to the Author

Reviewer #1: General Main Issues:

1. Comparing treated patients to untreated patients may be biased for several reasons. Most importantly, those with treatment differ in the severity of the symptoms and disease that lead to confounding by the indication, etc.

2. Very small sample size limits the generalizability of the study.

3. Insufficient/inappropriate variable use in the Mortality analyses (more details in the method section).

4. Lack of any sensitivity analyses to mitigate unmeasured confounding.

1. Abstract:

Overall, the abstract is structured and informative about the study’s objectives and results.

2. Introduction:

• In general, it would be beneficial if the authors discuss the result of previous studies on the association of pimavanserin with mortality and how their study is different and bring more insights.

• Lines 6-7. Authors included pimavanserin as an antipsychotic with a “low affinity for dopaminergic D2 receptors”, although it does not have an affinity, as explained later in the abstract.

3. Method:

Line5. It would be helpful if the authors explained why they did exclude the patients with primary psychiatric diagnoses or atypical parkinsonism.

Line7. It is unclear if the authors restricted the cohort to new users of pimavanserin and quetiapine and how they defined the “combination” class. Did the “combination” class include only patients with concomitant use of pimavanserin and quetiapine or also those with previous exposure and discontinuation? New users may be at higher risk of acute events than prevalent users (depletion of susceptibles). Authors should include more information regarding how they classified the treated group.

Line13. There is no explanation why the authors included “levodopa equivalent daily dose.” Did they consider it as a proxy for the severity of parkinsonism? Medication is not a way to assess disease severity in PDP. There are forms of PD that don’t respond to medication (and thus, levodopa might not be used). Patients with severe hallucinations/psychosis symptoms may use lower doses of levodopa.

Line15. Include more information on “common general medical conditions” definitions.

Line16. How will laboratory values be helpful to this study? If they are already included “common general medical conditions.”

Line17. By including the variable “at any time during the study period,” how did they handle the time-varying confounders?

3.1. Statistical Analyses:

Line7. Duration of exposure is a significant variable that has been overlooked in the regression model for mortality rate.

4. Result:

I would suggest authors refer to the related table at the end of each paragraph. The order of the result section is very confusing, and it is not aligned with the order in the method section and the study's objective. I would suggest authors to reorganize this section.

Line1. The numbers are reported without any percentage.

5. Discussion:

Overall, the authors did a good job with the discussion.

Reviewer #2: this is a nicely written and useful study. However, there are some questions that require answers and some modifications needed. You need to discuss how the patients classified as PDP were diagnosed. You list no criteria. You fail to state who made the dx of PD or PDP. I assume you have data on less than half the subjects because these data were not routinely obtained, which is fine, but bespeaks the weakness of your data. Since your subjects were identified via a visit to any UCSD facility, it is quite possible that some patient were dxd with PD by a PA somewhere, given a dx of PD and carries that dx into a UCSD emergency room, which is then engraved in stone forevermore. Similarly for PD psychosis.

In some sense this may not matter since the real goal here is to see whether pim or quetiapine increase mortality. There is a major oversight in your discussion of weaknesses, which is that you have no idea of how the various antipsychotics were chosen. presumably enough different providers were involved that it may not matter, but it could be that only the severely psychotic patients got quetiapine, or only the ones with poor sleeping. The study looked only at quetiapine vs pim, but other similar studies have included the other antipsychotics used, such as olanzapine and risperidone. Why were they not included?

Table 1 should be labelled as data obtained at admission to the study (first assessed visit).

6. PLOS authors have the option to publish the peer review history of their article (what does this mean?). If published, this will include your full peer review and any attached files.

Reviewer #1: No

Reviewer #2: No

---

## [Author Response · Author response to Decision Letter 0]

6 Nov 2022

Nov. 3, 2022

We thank the PLOS ONE reviewers for their time reviewing and thoughtful feedback for our manuscript entitled, “Assessing the risks of treatment in Parkinson disease psychosis: an in-depth analysis”, which has strengthened our paper. We have provided point-by-point responses to the reviewers’ comments below.

Reviewer #1:

General Main Issues:

1. Comparing treated patients to untreated patients may be biased for several reasons. Most importantly, those with treatment differ in the severity of the symptoms and disease that lead to confounding by the indication, etc.

Response: We acknowledge that there may be inherent biases in comparing treated to untreated patients. We have added the following sentences to acknowledge this issue in the limitations section (also in response to Reviewer 2’s comment #4):

“…we could not determine why one antipsychotic agent was chosen over another. Thus, we acknowledge that between-group differences may exist in psychosis severity or symptoms that could affect whether antipsychotic medication was prescribed, or which agent was prescribed.”

2. Very small sample size limits the generalizability of the study.

Response: We agree that the relatively small sample size limits generalizability of the study and have added the following sentence to the Discussion limitations paragraph: 

“We acknowledge the small sample size as a limitation to the generalizability of our findings.”

3. Insufficient/inappropriate variable use in the Mortality analyses (more details in the method section).

Response: In our current analysis, we have included age, sex, last visit LEDD, and dementia diagnosis as pre-specified covariates. These were chosen based on literature review given that dementia has a strong association with increased mortality in this cohort. The reviewer’s thoughtful comment regarding LEDD as a covariate is addressed in detail below. 

4. Lack of any sensitivity analyses to mitigate unmeasured confounding.

Response: Unfortunately, given the study design, information about age of disease onset, and motor severity was not available for all patients, so although these clinical factors have been associated with increased mortality in PD (see: Xu et al., Acta Neurologica Scandinavica 2014; Lo et al., Archives of Neurology, 2009), we were unable to include them as covariates. Due to our relatively small sample size – a limitation that we have added to our Discussions paragraph – we limited the number of covariates included to the ones that were most clinically relevant.

2. Introduction: 

In general, it would be beneficial if the authors discuss the result of previous studies on the association of pimavanserin with mortality and how their study is different and bring more insights.

Response: We have added the following sentences regarding previous studies about pimavanserin’s association with mortality in the introduction, and the new information that our study adds:

“Larger observational cohort studies have found varying results regarding pimavanserin’s association with mortality in PD. One study demonstrated among people not residing in long-term care facilities, pimavanserin users had decreased mortality compared to users of other antipsychotics [10]. However, another study found among residents of long-term care facilities that pimavanserin users had higher mortality compared to non-users [11].” We also clarified that in this study, we had the ability to perform individual chart review to extract details about our population.

We further elaborate on this topic in the Discussion.

• Lines 6-7. Authors included pimavanserin as an antipsychotic with a “low affinity for dopaminergic D2 receptors”, although it does not have an affinity, as explained later in the abstract.

Response: We thank the reviewer for noticing this inconsistency and have corrected this. The sentence now reads: 

“Direct PDP treatment is limited to the few antipsychotic medications that have low affinity for dopaminergic D2 receptors to avoid worsening parkinsonian symptoms; these include quetiapine and clozapine, which have traditionally been used to treat PDP.”

3. Methods:

Line 5. It would be helpful if the authors explained why they did exclude the patients with primary psychiatric diagnoses or atypical parkinsonism.

Response: We have clarified by adding the following sentences:

“We excluded patients with primary psychiatric diagnoses (including bipolar disorder, schizophrenia, schizotypal disorder, and depression with psychotic features), since these “may have drug-induced or tardive parkinsonism related to antipsychotic medication use. We excluded patients with atypical parkinsonism (e.g., multiple system atrophy, progressive supranuclear palsy, drug-induced parkinsonism, vascular parkinsonism), since pimavanserin is only FDA-approved for use in people with PDP.”

Line 7. It is unclear if the authors restricted the cohort to new users of pimavanserin and quetiapine and how they defined the “combination” class. Did the “combination” class include only patients with concomitant use of pimavanserin and quetiapine or also those with previous exposure and discontinuation? New users may be at higher risk of acute events than prevalent users (depletion of susceptibles). Authors should include more information regarding how they classified the treated group.

Response: We thank the reviewer for this comment. We have clarified in the paragraph regarding medication exposure by adding the following:

“The combination group included patients that had exposure to both pimavanserin and quetiapine during the study period, but these two agents were not necessarily taken at the same time. Patients who may have been exposed to antipsychotic medications and discontinued them before the study period were included in the untreated group.”

We have acknowledged the lack of information regarding medication exposure duration as a limitation in this study.

Line13. There is no explanation why the authors included “levodopa equivalent daily dose.” Did they consider it as a proxy for the severity of parkinsonism? Medication is not a way to assess disease severity in PDP. There are forms of PD that don’t respond to medication (and thus, levodopa might not be used). Patients with severe hallucinations/psychosis symptoms may use lower doses of levodopa.

Response: We have clarified that LEDD was included as a covariate since dopaminergic therapy may increase risk of psychosis, and have added the following supportive references:

Forsaa EB, Larsen JP, Wentzel-Larsen T, Goetz CG, Stebbins GT, Aarsland D, Alves G. A 12-year population-based study of psychosis in Parkinson disease. Archives of Neurology. 2010 Aug 1;67(8):996-1001.

Zhu, Kangdi, et al. "Risk factors for hallucinations in Parkinson's disease: results from a large prospective cohort study." Movement Disorders 28.6 (2013): 755-762.

Line 15. Include more information on “common general medical conditions” definitions.

Response: We have clarified the medical conditions that were searched using ICD-10 codes included cardiovascular disease, chronic kidney disease, diabetes mellitus type 2, hypertension, hyperlipidemia, and hypercoagulability.” 

Line 16. How will laboratory values be helpful to this study? If they are already included “common general medical conditions.”

Response: We have clarified that we added laboratory values as an objective measure of these conditions. 

Line 17. By including the variable “at any time during the study period,” how did they handle the time-varying confounders?

We have clarified that medical exposure/condition variables were defined as any exposure during the study period, and we have rephrased the manuscript accordingly. For the continuous covariates, we examined the average effect over the study period. While time-varying covariates are usually considered in longitudinal data analysis or survival data analysis, but due to the limitation of our study design, it is not feasible to perform survival analysis and we examined mortality as a binary outcome instead. We acknowledged in the Limitations that we were unable to perform this kind of analysis. 

3.1. Statistical Analyses:

Line7. Duration of exposure is a significant variable that has been overlooked in the regression model for mortality rate.

Response: We agree with the reviewer that exposure duration is an important variable. However, since the study design was a retrospective review, unfortunately, there was no way to extract this information from the medical records. We acknowledge this is a major limitation to the study. 

4. Results:

I would suggest authors refer to the related table at the end of each paragraph. The order of the result section is very confusing, and it is not aligned with the order in the method section and the study's objective. I would suggest authors to reorganize this section.

Response: We thank the reviewer for the feedback. We have reorganized the methods and results section to be more congruent and clearer and placed the tables at the end of each paragraph. 

Line1. The numbers are reported without any percentage.

Response: We have added the percentages to the first paragraph of the Results section, which now reads,

“Using our inclusion/exclusion criteria, the sample included 2,994 PD patients – 352 (11.8%) with psychosis. Of these 352 PDP patients, 66 (18.8%) were untreated (did not receive antipsychotics), 34 (9.7%) received pimavanserin, 147 (41.8%) received quetiapine in the outpatient setting, and 68 (19.3%) received combination therapy, thus 315 patients were included in the analyses.”

Reviewer #2: This is a nicely written and useful study. However, there are some questions that require answers and some modifications needed. 

1. You need to discuss how the patients classified as PDP were diagnosed. 

You list no criteria. 

Response: We thank the reviewer for raising this point. We have clarified that patients were diagnosed with PD and psychosis based on ICD-10 code. For patients who were prescribed antipsychotic medications, individual chart review was performed to ascertain that the medication was prescribed for psychosis (i.e., rather than for sleep or mood) and those without psychosis were excluded.

2. You fail to state who made the dx of PD or PDP. I assume you have data on less than half the subjects because these data were not routinely obtained, which is fine, but bespeaks the weakness of your data. Since your subjects were identified via a visit to any UCSD facility, it is quite possible that some patient were dxd with PD by a PA somewhere, given a dx of PD and carries that dx into a UCSD emergency room, which is then engraved in stone forevermore. Similarly for PD psychosis. In some sense this may not matter since the real goal here is to see whether pim or quetiapine increase mortality.

Response: Unfortunately, data regarding the provider who made the diagnosis of PD or PDP was not available for all subjects given the retrospective nature of the study. Many patients - but not all - were treated by a movement disorders neurologist. We acknowledge this is a limitation to our study and have added this point to our limitation paragraph. However, as the reviewer mentioned, this limitation would unlikely differ between the treated and untreated groups. 

4. There is a major oversight in your discussion of weaknesses, which is that you have no idea of how the various antipsychotics were chosen. presumably enough different providers were involved that it may not matter, but it could be that only the severely psychotic patients got quetiapine, or only the ones with poor sleeping. 

Response: We thank the reviewer for bringing this weakness to our attention. We have added this as a limitation to our Discussion. 

5. The study looked only at quetiapine vs pim, but other similar studies have included the other antipsychotics used, such as olanzapine and risperidone. Why were they not included?

Response: The sample size of PDP patients taking other antipsychotics was small (n=9 treated with clozapine and other antipsychotics during the study period) - though we did not include these other agents as monotherapy in our search query). We have added to the manuscript: 

“We did not include other atypical antipsychotic medications, e.g., risperidone or olanzapine, in our search query since these and other antipsychotics with D2 dopaminergic blocking mechanism of action are generally avoided in PD given their propensity to exacerbate parkinsonian symptoms.”

Olanzapine specifically is not recommended for treatment of PDP by the International Parkinson and Movement Disorders Society evidenced-based treatment review due to “unacceptable risk of motor deterioration” (Seppi et al. 2019).

6. Table 1 should be labeled as data obtained at admission to the study (first assessed visit)

Response: We have clarified in the Table 1 title and in the manuscript text that this data is the baseline information from the first assessed visit. 

Thank you,

Fatta B. Nahab, M.D., FAAN

Department of Neurosciences

University of California San Diego 

9500 Gilman Drive, Mailcode: 0886

La Jolla, CA 92093

fnahab@health.ucsd.edu

---

## [Editor Report · Decision Letter 1]

14 Nov 2022

Assessing the risks of treatment in Parkinson disease psychosis: an in-depth analysis

PONE-D-22-22035R1

Dear Dr. Fatta B. Nahab,

We’re pleased to inform you that your manuscript has been judged scientifically suitable for publication and will be formally accepted for publication once it meets all outstanding technical requirements.

Kind regards,

Antonina Luca, MD, PhD

Academic Editor

PLOS ONE

---

## [Editor Report · Acceptance letter]

21 Nov 2022

PONE-D-22-22035R1 

Assessing the risks of treatment in Parkinson disease psychosis: an in-depth analysis 

Dear Dr. Nahab:

I'm pleased to inform you that your manuscript has been deemed suitable for publication in PLOS ONE. Congratulations! Your manuscript is now with our production department. 

Kind regards, 

on behalf of

Dr. Antonina Luca 

Academic Editor

PLOS ONE